# Generating Synthetic Health Sensor Data for Privacy-Preserving Wearable Stress Detection

**DOI:** 10.3390/s24103052

**Published:** 2024-05-11

**Authors:** Lucas Lange, Nils Wenzlitschke, Erhard Rahm

**Affiliations:** ScaDS.AI Dresden/Leipzig, Leipzig University, Augustusplatz 10, 04109 Leipzig, Germany; nw20hewo@studserv.uni-leipzig.de (N.W.); rahm@informatik.uni-leipzig.de (E.R.)

**Keywords:** generative adversarial network, stress recognition, privacy-preserving machine learning, differential privacy, smartwatch, time series, physiological sensor data, synthetic data, smart health

## Abstract

Smartwatch health sensor data are increasingly utilized in smart health applications and patient monitoring, including stress detection. However, such medical data often comprise sensitive personal information and are resource-intensive to acquire for research purposes. In response to this challenge, we introduce the privacy-aware synthetization of multi-sensor smartwatch health readings related to moments of stress, employing Generative Adversarial Networks (GANs) and Differential Privacy (DP) safeguards. Our method not only protects patient information but also enhances data availability for research. To ensure its usefulness, we test synthetic data from multiple GANs and employ different data enhancement strategies on an actual stress detection task. Our GAN-based augmentation methods demonstrate significant improvements in model performance, with private DP training scenarios observing an 11.90–15.48% increase in F1-score, while non-private training scenarios still see a 0.45% boost. These results underline the potential of differentially private synthetic data in optimizing utility–privacy trade-offs, especially with the limited availability of real training samples. Through rigorous quality assessments, we confirm the integrity and plausibility of our synthetic data, which, however, are significantly impacted when increasing privacy requirements.

## 1. Introduction

Healthcare applications see an ever-growing need for high-quality medical data in abundant amounts. In particular, the uprising of smart health services can provide valuable insights into individual health conditions and personalized remedy recommendations. For example, solutions for detecting stress from physiological measurements of wearable devices receive attention from academic [1,2,3] and industrial [4,5,6] communities alike.

However, each entry in a medical dataset often contains detailed information about an individual’s health status, making it highly sensitive and leading to various anonymization techniques [7,8,9]. Still, the risk of re-identification persists, as current methods can successfully identify individuals based solely on their health signal data [10,11,12,13]. These threats ultimately lead to complex ethical and privacy requirements, complicating the collection and access to sufficient patient data for real-world research [14,15].

Regarding patient privacy, training machine learning models under the constraints of Differential Privacy (DP) [16] provides a robust and verifiable privacy guarantee. This approach ensures the secure handling of sensitive data and effectively mitigates the risk of potential attacks when these models are deployed in operational settings.

To address the limitations related to data availability, one effective strategy is the synthesis of data points, often achieved through techniques like Generative Adversarial Networks (GANs) [17]. GANs enable the development of models that capture the statistical distribution of a given dataset and subsequently leverage this knowledge to generate new synthetic data samples that adhere to the same foundational principles. In addition, we can directly integrate the privacy assurances of DP into our GAN training process, enabling the direct creation of a privacy-preserving generation model. This ensures that the synthetically generated images offer and maintain privacy guarantees [18].

In this work, we train both non-private GAN and private DP-GAN models to generate new time-series data needed for smartwatch stress detection. Existing datasets for stress detection are small and can benefit from augmentation, especially when considering difficulties in the private training of detection models using DP [19]. We present and evaluate multiple strategies for incorporating both non-private and private synthetic data to enhance the utility–privacy trade-off introduced by DP. Through this augmentation, our aim is to optimize the performance of privacy-preserving models in a scenario where we are constrained by limited amounts of real data.

Our contributions are as follows:We achieve data generation models based on GANs that produce synthetic multimodal time-series sequences corresponding to available smartwatch health sensors. Each data point presents a moment of stress or non-stress and is labeled accordingly.Our models generate realistic data that are close to the original distribution, allowing us to effectively expand or replace publicly available, albeit limited, data collections for stress detection while keeping their characteristics and offering privacy guarantees.With our solutions for training stress detection models with synthetic data, we are able to improve on state-of-the-art results. Our private synthetic data generators for training DP-conform classifiers help us in applying DP with much better utility–privacy trade-offs and lead to higher performance than before. We give a quick overview regarding the improvements over related work in Table 1.Our approach enables applications for stress detection via smartwatches while safeguarding user privacy. By incorporating DP, we ensure that the generated health data can be leveraged freely, circumventing privacy concerns of basic anonymization. This facilitates the development and deployment of accurate models across diverse user groups and enhances research capabilities through increased data availability.

In Section 2, we briefly review relevant basic knowledge and concepts before focusing on existing related work regarding synthetic health data and stress detection in Section 3. Section 4 presents an overview of our methodology, describes our experiments, and gives reference to the environment for our implementations. The outcome of our experiments is then detailed and evaluated in Section 5. Section 6 is centered around discussing the implications of our results and determining the actual best strategies from different perspectives, as well as their possible limitations. Finally, in Section 7, we provide both a concise summary of our key findings and an outlook on future work.

## 2. Background

The following section introduces some of the fundamental concepts used in this work.

### 2.1. Stress Detection from Physiological Measurements

A key factor in mobile stress detection systems is the availability and processing of these sensor readings, which also leads to the question of which sensors we are able to measure using wearables and how relevant each sensor might be in classifying stress correctly. In wrist-worn wearable devices commonly used for stress-related research purposes like the Empatica E4 [23] we find three-way acceleration (ACC[x,y,z]), electrodermal activity (EDA) also known as galvanic skin response (GSR), skin temperature (TEMP), and blood volume pressure (BVP), which also doubles as an indicator of heart rate (HR). Especially EDA is known as a key instrument for identifying moments of stress, while the electroencephalogram (EEG) also gives strong indications but has less availability in continuous wrist readings [1].

There are numerous reactions of the human body when answering situations of stress or amusement. Giannakakis et al. [1] give a comprehensive list of studies and separate measurable biosignals related to stress into two categories: physiological (EEG, EDA) and physical measures (respiratory rate, speech, skin temperature, pupil size, eye activity). Some of the found correlations are, e.g., TEMP: high in non-stress and low in stress, EDA: low in non-stress and high in stress, and BVP, which has a higher frequency in the stress state than in the non-stress state.

### 2.2. Generative Adversarial Network

The Generative Adversarial Network (GAN), introduced by Goodfellow et al. [17], is a type of machine learning framework that trains two neural networks concurrently. It consists of a generative model, denoted as *G*, which learns the data distribution, and a discriminative model, denoted as *D*, which estimates the likelihood of a sample coming from the dataset versus *G*. The architecture of the original GAN model is depicted in Figure 1. The objective of the generator *G* is to generate realistic data samples. The discriminator *D* then receives both the synthetically generated samples and the real samples and classifies each sample as either real or fake. The generator learns indirectly through its interaction with the discriminator, as it does not have direct access to the real samples. The discriminator generates an error signal based on the ground truth of whether the sample came from the real dataset or the generator. This error signal is then used to train the generator via the discriminator, leading to the production of improved synthetic samples. Consequently, *G* is trained to maximize the probability of *D* making an error, and the training process takes the form of a min–max game, where the error of *G* should be minimized and the error of *D* maximized.

### 2.3. Differential Privacy

Differential Privacy (DP), as defined by Dwork [16], is a mathematical approach to privacy. It guarantees that the addition or removal of a single data point in a dataset does not significantly impact the results of statistical computations on that dataset. This is achieved by adding a certain level of noise to the computations, which obscures the effect of individual data points but still allows for meaningful analysis of the dataset.

In technical terms, we say that an algorithm *A* that works on a set *S* is (ε,δ)-differentially private if the following condition is met for any pair of datasets *D* and $D^{\prime }$ that differ by just one data point:(1)$$Pr\left[A\left(D\right)\in S\right]\leq e^{\varepsilon }Pr\left[A\left(D^{\prime }\right)\in S\right]+\delta .$$

The ε parameter, often referred to as the privacy budget, quantifies the degree of privacy offered by the mechanism. It controls the amount of random noise added to the computations on the dataset. A smaller ε value provides stronger privacy protections but reduces the utility of the data due to the increased noise. In general, an ε value less than or equal to one is considered to provide strong privacy protections [24,25,26].

### 2.4. Differentially Private Stochastic Gradient Descent

Differentially Private Stochastic Gradient Descent (DP-SGD), as introduced by Abadi et al. [27], is a modification of the traditional stochastic gradient descent optimization method that incorporates DP principles. The key idea behind DP-SGD is the introduction of a controlled level of noise to the gradient calculations performed on each data mini-batch during the model training phase. The magnitude of the noise introduced is governed by setting the privacy budget parameter, denoted as ε, which serves as a measure of the level of DP protection offered. The process of setting the value of the ε parameter can be complex, and it requires careful consideration and adjustment of the noise level to strike a balance between privacy protection and data utility.

## 3. Related Work

This section gives reviews of the other literature in the associated fields of research.

### 3.1. Synthetic Data for Stress Detection

Ehrhart et al. [28] successfully introduced a GAN approach for a very similar use case, but without considering privacy. In their study, they collected Empatica E4 wristband sensor data from 35 individuals in baseline neutral situations and when inducing stress through air horn sounds. They then trained a Conditional GAN (CGAN) architecture to generate realistic EDA and TEMP signal data. These data are then used to augment the existing data basis and improve their stress detection results. Due to data protection laws, we are not able to use their dataset for our private approach and we are instead limited to using the publicly available but smaller WESAD dataset [20] with 15 participants, which was also collected using the E4 wristband. In contrast to Ehrhart et al. [28], we focus on generating the full range of the available six sensor modalities (ACC[x,y,z], EDA, TEMP, BVP), while they only focused on two of them in their GAN model. We build on their valuable research by using data available to the public, including more sensor modalities, and furthermore, by giving a new perspective on approaches for privacy preservation in stress detection from such data.

### 3.2. Privacy of Synthetic Data

The relevance of privacy methods might seem contradictory at first since the approach of using synthetic data instead of real data itself already seems to hide the original information found in the data source. Contrary to this intuition, we find that synthetic data can still provide exploitable information on the dataset it is meant to resemble, which is especially true for data generated by GANs [29]. This contradiction is less surprising on second thought; since the goal of synthetic data is to closely follow the distribution of real data, there has to be some inherent information on its distributional qualities hidden inside the synthetic fakes. Another factor making GAN data vulnerable is the general nature of machine learning, where models tend to overly memorize their training data and, as with all models, GANs will have the same difficulties escaping this paradigm [25]. Xie et al. [18] give a solution to these privacy concerns in the form of their DP-GAN model, which is a general GAN model, where the generator is trained using the widespread DP-SGD algorithm to attain private models that guarantee DP. Thanks to this modular design, the DP-GAN approach can be applied to different underlying GAN models and for any given data, like a possible DP-CGAN architecture presented by Torkzadehmahani et al. [30].

### 3.3. Stress Detection on WESAD Dataset

There are multiple recent works in smartwatch stress detection that are evaluated on the WESAD dataset introduced by Schmidt et al. [20], which is a common choice inside the research field. We list the relevant results from Table 1 but filter them to only include models based on wrist-based wearable devices that classify samples into *stress* and *non-stress*. The Convolutional Neural Network (CNN) model [22] delivers the best performance in the non-private setting at ε=∞, outperforming, amongst others, the Random Forest (RF) [20] and Linear Discriminant Analysis (LDA) [21] solutions. The Time-Series Classification Transformer (TSCT) approach [19] also stays slightly behind, but on the other hand, showed to be the only related work employing DP for this task. Taking these numbers as our reference for the utility–privacy trade-off suggests that we should expect a substantial draw-down in performance when aiming for any of these privacy guarantees. However, when comparing our best results using synthetic data in Table 1, we improve on both the non-private and private settings. The utility–privacy trade-off improves significantly, especially at ε=0.1, which is a very strict guarantee.

## 4. Methodology

In this part, we detail the different methods and settings for our experiments. A general overview is given in Figure 2, while each process and each presented part are further described in the following section.

### 4.1. Environment

On the software side, we employ Python 3.8 as our programming language and utilize the Tensorflow framework for our machine-learning models. The accompanying Tensorflow Privacy library provides the relevant DP-SGD training implementations. Our hardware configuration for the experiments comprises machines with 32 GB of RAM and an NVIDIA GeForce RTX 2080 Ti graphics card. We further set the random seed to 42.

### 4.2. Dataset Description

Our proposed method is examined on the openly accessible multimodal WESAD dataset [20], a frequently utilized dataset for stress detection. The dataset is composed of 15 healthy participants (12 males, 3 females), each with approximately 36 min of health data recorded during a laboratory study. Throughout this time, data are continuously and concurrently collected from a wrist-worn and a chest-worn device, both providing multiple modalities as time-series data. We limit our consideration to signals obtainable from wrist-worn wearables, such as smartwatches—specifically, the Empatica E4 device used in the dataset. The wristband provides six modalities at varying sampling frequencies: blood volume pulse (BVP), electrodermal activity (EDA), body temperature (TEMP), and three-axis acceleration (ACC[x,y,z]). The dataset records three pertinent affective states: neutral, stress, and amusement. Our focus is on binary classification, distinguishing stress from non-stress, which merges the neutral and amusement classes. Ultimately, we find the data comprise approximately 30% stress and 70% non-stress instances.

### 4.3. Data Preparation

Transforming time-series signal data to match the expected input format requires several pre-processing steps and is a crucial step in achieving good models. For our approach, we adopt the process of Gil-Martin et al. [22] in many points. We, however, change some key transformations to better accommodate the data to our setting and stop at 60 s windows since we want to feed them into our GANs instead of their CNN model. Our process can be divided into four general steps.

First, since the Empatica E4 signal modalities are recorded at different sampling rates due to technical implementations, they need to be resampled to a unified sampling rate. We further need to align these sampled data points to ensure that for each point of time in the time series, there is a corresponding entry for all signals. To achieve this, signal data are downsampled to a consistent sampling rate of 1 Hz using the Fourier method. Despite the reduction in original data points, most of the crucial non-stress/stress dynamics are still captured after the Fourier transformation process, while model training is greatly accelerated by reducing the number of samples per second. An additional result is the smoothing of the signals, which helps the GAN in learning the important overall trends without smaller fluctuations present due to higher sampling rates.

In the second step, we adjust the labels by combining *neutral* and *amusement* into the common *non-stress* label. In addition to these data, we only keep the *stress* part of the dataset. This reduction in labels is mainly due to the fact that we want to enhance binary stress detection that only distinguishes between moments of stress and moments without stress. However, only keeping neutral data would underestimate the importance of differentiating the amusement phase from the stress phase since there is an overlap in signal characteristics, such as BVP or ACC, for amusement and stress [20]. After the first and this relabeling step, we obtain an intermediate result of 23,186 non-stress- and 9966 stress-labeled seconds.

Thirdly, we normalize the signals using a min–max normalization in the range of [0,1] to eliminate the differences in scale among the modalities while still capturing their relationships. In addition, the normalization has a great impact on the subsequent training process, as it helps the model to converge faster, thus shortening the time to learn an optimal weight distribution.

Given that the dataset consists of about 36 min sessions per subject, in our fourth and final step, we divide these long sessions into smaller time frames to pose as input windows for our models. We transform each into 60-s long windows but additionally, as described by Dzieżyc et al. [31], we introduce a sliding window effect of 30 s. This means instead of neatly dividing into 60-s windows, we instead create a 60-s window after every 30 s of the data stream. These additional intermediate windows fill the gap between clean aligned 60-s windows by overlapping with the previous window by 30 s and the next window by 30 s, providing more contextual information by capturing the correlated time series between individual windows. Additionally, sliding windows increase the amount of data points available for subsequent training. We opt for 30-s windows over shorter ones to limit the repeating inclusion of unique data points, which would escalate the amount of DP noise with increased sampling frequency, as detailed in Section 4.8. A lower amount of overlapping windows ensures manageable DP noise, while still giving more samples. To assign a label for a window, we determine the majority class in the given 60-s frame. Finally, we concatenate the 60-s windows and their associated class labels from all subjects into a final training dataset.

An example of pre-processed data is given in Figure 3, where we show the graphs for Subject ID4 from the WESAD dataset after the first three processing steps. The orange line represents the associated label for each signal plot and is given as 0 for non-stress and 1 for stress. We can already spot certain differences between the two states in relation to the signal curves simply when looking at the given plots.

### 4.4. Generative Models

After transforming our signal data to a suitable and consistent input format, it is important to determine the proper model architecture for the given data characteristics. Compared to the original GAN architecture [17], we face three main challenges:*Time-series data*: Instead of singular and individual input samples, we find continuous time-dependent data recorded over a specific time interval. Further, each data point is correlated to the rest of the sequence before and after it.*Multimodal signal data*: For each point in time, we find not a single sample but one each for all of our six signal modalities. Artificially generating this multimodality is further complicated by the fact that the modalities correlate to each other and to their labels.*Class labels*: Each sample also has a corresponding class label as stress or non-stress. This is solvable with standard GANs by training a separate GAN for each class, like when using the Time-series GAN (TimeGAN) [32]. However, with such individual models, some correlation between label and signal data might be lost.

Based on these data characteristics and resulting challenges, we have selected the following three GAN architectures that address these criteria in different ways.

#### 4.4.1. Conditional GAN

The Conditional GAN (CGAN) architecture was first introduced by Mirza and Osindero [33]. Here, both the generator and the discriminator receive additional auxiliary input information, such as a class label, with each sample. This means that, in addition to solely generating synthetic samples, the CGAN is able to learn and output the corresponding labels for synthetic samples, effectively allowing the synthetization of labeled multimodal data. For our time-series CGAN variant, we mainly adopt the architecture and approach from the related work by Ehrhart et al. [28]. They also evaluated the CGAN against the TimeGAN and determined that the TimeGAN’s generative performance was inferior for our specific task. Consequently, we chose to exclude the TimeGAN from our evaluation, given its inferiority to the CGAN. The used CGAN architecture is based on the LSTM-CGAN [34] but is expanded by a diversity term to stabilize training and an FCN discriminator model with convolutional layers. We instead rely on an LSTM discriminator by stacking two LSTM layers, which performs better in our scenario [35]. As hyperparameters, we choose the diversity term λ=8 and employ an Adam [36] optimizer with a learning rate of $2\times 10^{-4}$. We further pick 64 for the batch size and train for 1600 epochs. We derived these values from hyperparameter tuning.

#### 4.4.2. DoppelGANger GAN

The other architecture considered is the DoppelGANger GAN (DGAN) by Lin et al. [37]. Like the CGAN, the DGAN uses LSTMs to capture relationships inside the time-series data. Thanks to a new architectural element, the DGAN is able to include multiple generators in its training process. The goal is to decouple the conditional generation part from the time-series generation. They thus include separate generators for auxiliary metadata, like labels, and continuous measurements. In the same vein, they use an auxiliary discriminator in addition to the standard discriminator, which exclusively judges the correctness of metadata outputs. To address mode collapse problems, they further introduce a third generator, which again treats the min and max of signal values as metadata. By combining these techniques, Lin et al. [37] try to incorporate the relationships between the many different attributes. This approach also offers the advantage that a trained model can be further refined, and by flexibly changing the metadata, can generate synthetic data for a different use case. In terms of hyperparameters, we choose a learning rate of $1\times 10^{-3}$ and train for 10,000 epochs with the number of training samples as the batch size.

#### 4.4.3. DP-CGAN

Our private DP-GAN architecture of choice is the DP-CGAN, which was already used by Torkzadehmahani et al. [30], without our focus on time-series data. Through the multiple generators and discriminator parts, the DGAN has a harder time complying with private training, which is why we stayed with the CGAN for private training that performed well in the initial tests. To incorporate our task into the architecture, we take the CGAN part from Ehrhart et al. [28] and make it private using DP-SGD. More specifically, we use the DP-Adam optimizer, which is an Adam variant of DP-SGD. For privatizing the CGAN architecture, we draw on the DP-GAN ideas by both Xie et al. [18] and Liu et al. [38]. Both approaches introduce the concept of securing a DP guarantee for GANs via applying noise to the gradients through the optimizer during training. During GAN training, the generator only reacts to the feedback received from the discriminator, while the discriminator is the part that accesses real data for calculating the loss function [38]. From this, we can determine that just the discriminator needs to implement noise injection when seeing real samples to hide their influence. Thus, only the discriminator needs to switch to the DP optimizer and the generator can keep its standard training procedure. The hyperparameters of DP-CGAN training are described in Section 4.8, where we focus on the necessary information for implementing the private training.

### 4.5. Synthetic Data Quality Evaluation

Under the term of data quality, we unite the visual and statistical evaluation methods for our synthetic data. We use the following four strategies to obtain a good understanding of the achieved diversity and fidelity provided by our GANs:*Principal Component Analysis (PCA)* [39]. As a statistical technique for simplifying and visualizing a dataset, PCA converts many correlated statistical variables into principal components to reduce the dimensional space. Generally, PCA is able to identify the principal components that identify the data while preserving their coarser structure. We restrict our analysis to calculating the first two PCs, which is a feasible representation since the major PCs capture most of the variance.*t-Distributed Stochastic Neighbor Embedding (t-SNE)* [40]. Another method for visualizing high-dimensional data is using t-SNE. Each data point is assigned a position in a two-dimensional space. This reduces the dimension while maintaining significant variance. Unlike PCA, it is less qualified at preserving the location of distant points, but can better represent the equality between nearby points.*Signal correlation and distribution*. To validate the relationship between signal modalities and to their respective labels, we analyze the strength of the Pearson correlation coefficients [41] found inside the data. A successful GAN model should be able to output synthetic data with a similar correlation as the original training data. Even though correlation does not imply causation, the correlation between labels and signals can be essential to train classification models. Additionally, we calculate the corresponding *p*-values (probability values) [42] to our correlation coefficients to analyze if our findings are statistically significant. As a further analysis, we also take a look at the actual distribution of signal values to see if the GANs are able to replicate these statistics.*Classifier Two-Sample Test (C2ST)*. To evaluate whether the generated data are overall comparable to real WESAD data, we employ a C2ST mostly as described by Lopez-Paz and Oquab [43]. The C2ST uses a classification model that is trained on a portion of both real and synthetic data, with the task of differentiating between the two classes. Afterward, the model is fed with a test set that again consists of real and synthetic samples in equal amounts. Now, if the synthetic data are close to the real data, the classifier would have a hard time correctly labeling the different samples, leaving it with a low accuracy result. In an optimal case, the classifier would label all given test samples as real and thus only achieve 0.5 of accuracy. This test method allows us to see if the generated data are indistinguishable from real data for a trained classifier. For our C2ST model, we decided on a Naive Bayes approach.

### 4.6. Use Case Specific Evaluation

We test the usefulness of our generated data in an actual stress detection task for classifying stress and non-stress data. The task is based on the WESAD dataset and follows an evaluation scheme using Leave One Subject Out (LOSO) cross-validation. In standard machine-learning evaluation, we would split the subjects from the WESAD dataset into distinct train and test sets. In this scenario, we would only test on the selected subjects, and these would also be excluded from training. In the LOSO format, we instead train 15 different models, one for each subject in the WESAD dataset. A training run uses 14 of the 15 subjects from the WESAD dataset as the training data and the 15th subject as the test set for evaluation. Thereby, when cycling through the whole dataset using this strategy, every subject constitutes the test set once and is included in the training for the 14 other runs. This allows us to evaluate the classification results for each subject. For the final result, all 15 test set results are averaged into one score, simulating an evaluation for all subjects. This process is also performed by the related work presented in Table 1.

To evaluate our synthetic data, we generate time-series sequences per GAN model with the size of an average subject of roughly 36 min in the WESAD dataset. We also conform to the same distribution of stress and non-stress with about 70% and 30%, respectively. By this, we want to generate comparable subject data that allow us to realistically augment or replace the original WESAD dataset with synthetic data. We can then evaluate the influence of additional subjects on the classification. The synthetic subjects are included in each training round of the LOSO evaluation but the test sets are only based on the original 15 subjects to obtain comparable and consistent results. The GANs are also part of the LOSO procedure, which means the subject that currently provides the test set is omitted from their training. Finally, each full LOSO evaluation run is performed 10 times to better account for randomness and fluctuations from the GAN data, classifier training, and DP noise. The results are then again averaged into one final score.

For an evaluation metric, we use the F1-score over accuracy since it combines both precision and recall and shows the balance between these metrics. The F1-score gives their harmonic mean and is particularly useful for unbalanced datasets, such as the WESAD dataset with its minority label distribution for stress. Precision is defined as $Prec=\frac{TP}{TP+FP}$, while recall is $Rec=\frac{TP}{TP+FN}$, and the F1-score is then given as $F1=2\times \frac{\mathrm{Prec}\times \mathrm{Rec}}{\mathrm{Prec}+\mathrm{Rec}}$.

To improve the current state-of-the-art classification results using our synthetic data, we test the following two strategies in both non-private and private training scenarios:*Train Synthetic Test Real (TSTR).* The TSTR framework is commonly used in the synthetic data domain, which means that the classification model is trained on just the synthetic data and then evaluated on the real data for testing. We implement this concept by generating synthetic subject data in differing amounts, i.e., the number of subjects. We decide to first use the same size as the WESAD set of 15 subjects to simulate a synthetic replacement of the dataset. We then evaluate a larger synthetic set of 100 subjects. Complying with the LOSO method, the model is trained using the respective GAN model, leaving out the test subject on which it is then tested. The average overall subject results are then compared to the original WESAD LOSO result. Private TSTR models can use our already privatized DP-CGAN data in normal training.*Synthetic Data Augmentation (AUGM).* The AUGM strategy focuses on enlarging the original WESAD dataset with synthetic data. For each LOSO run of a WESAD subject, we combine the respective original training data and our LOSO-conform GAN data in differing amounts. As before in TSTR, we consider 15 and 100 synthetic subjects. Testing is also performed in the LOSO format. With this setup, we evaluate if adding more subjects, even though synthetic and of the same nature, helps the classification. Private training in this scenario takes the privatized DP-CGAN data but also has to consider the not-yet-private original WESAD data they are combined with. Therefore, the private AUGM models still undergo a DP-SGD training process to guarantee DP.

### 4.7. Stress Classifiers

In the following section, we present the tested classifier architectures and their needed pre-processing.

#### 4.7.1. Pre-Processing for Classification

After already pre-processing our WESAD data for GAN training, as described in Section 4.3, we now need the aforementioned further processing steps from Gil-Martin et al. [22] to transform our training data into the correct shape for posing as inputs to our classification models. The 60-s long windows from Section 4.3 are present in both the WESAD and synthetically generated data. The only difference between the two is that we do not apply the 30-s sliding window to the original WESAD data as we applied before for the GAN training.

In the next step, we want to convert each window into a frequency-dependent representation using the Fast Fourier Transformation (FFT). The FFT is an efficient algorithm for computing the Fourier transform, which transforms a time-dependent signal into the corresponding frequency components that constitute the original signal. This implies that these windows are converted into frequency spectra. However, before applying the FFT, we further partition the 60-s windows into additional subwindows of varying lengths based on the signal type. For these subwindows, we implement a sliding window of 0.25 s. The varying lengths of the subwindows are due to the distinct frequency spectrum characteristics of each signal type. We modify the subwindow length based on a signal’s frequency range to achieve a consistent spectrum shape comprising 210 frequency points.

Gil-Martin et al. [22] provide each signal’s frequency range and give the corresponding subwindow length as shown in Table 2. The subwindow lengths are chosen to always result in the desired 210 data points when multiplied by the frequency range upper bound, which will be the input size for the classification models. An important intermediate step for our GAN-generated data to avoid possible errors in dealing with missing frequencies in the higher ranges is to, in some cases, pad the FFT subwindows with additional zeroes to reach the desired 210 points. The frequency spectra are then averaged along all subwindows inside a 60-s window to finally obtain a single averaged spectrum representation with 210 frequency points to represent a 60-s window. We plot the spectrum results for the subwindows of a 60-s window in Figure 4a and show their final averaged spectrum representation in Figure 4b. Higher amplitudes are more present in the lower frequencies.

#### 4.7.2. Time-Series Classification Transformer

As our first classification model, we pick the Time-Series Classification Transformer (TSCT) from Lange et al. [19] that delivers the only comparison for related work in privacy-preserving stress detection, which is also described in Section 3. The model is, however, unable to reach the best state-of-the-art results for the non-private setting. In their work, the authors argue that the transformer model could drastically benefit from more training samples, like our synthetic data. In our implementation, we use class weights and train for 110 epochs with a batch size of 50 using the Adam optimizer at a $1\times 10^{-3}$ learning rate.

#### 4.7.3. Convolutional Neural Network

The Convolutional Neural Network (CNN) is the currently best-performing model in the non-private setting presented by Gil-Martin et al. [22]. For our approach, we also include their model in our evaluations to see if it keeps the top spot. We mostly keep the setup of the TSCT in terms of hyperparameters but train the CNN for just 10 epochs.

#### 4.7.4. Hybrid Convolutional Neural Network

As the final architecture, we consider a hybrid LSTM-CNN model, for which we take the same CNN architecture but add two Long Short-Term Memory (LSTM) layers of sizes 128 and 64 between the convolutional part and the dense layers. Through these additions, we want to combine the advantages of the state-of-the-art CNN and the ability to recognize spatial correlations in the time series from the LSTM. For the hyperparameters, we keep the same setup as for the standard CNN but increase the training time to 20 epochs.

### 4.8. Private Training

In this section, we go over the necessary steps and parameters to follow our privacy implementation. We first focus on the training of our private DP-CGANs and then follow with the private training of our classification models.

We want to evaluate three DP guarantees that represent different levels of privacy. The first has a budget of ε=10 and is a more relaxed although still private setting. The second and third options are significantly stricter in their guarantees, with a budget of ε=1 and ε=0.1. The budget of ε=1 is already considered strong in the literature [24,25,26], making the setting of ε=0.1 a very strict guarantee. Giving a less privacy budget leads to higher induced noise during training and therefore a higher utility loss. We want to test all three values to see how the models react to the different amounts of randomness and privacy.

#### 4.8.1. For Generative Models

We already described our private DP-CGAN models in Section 4.4 and now offer further details on how we choose the hyperparameters relevant to their private training. The induced noise at every training step needs to be calculated depending on the wanted DP guarantee and under the consideration of the training setup. We switch to a learning rate of 1e−3, set the epochs to 420, and take a batch size of 8, which is also our number of microbatches. Next, we determine the number of samples in the training dataset, which for us is the number of windows. By applying a 30-s sliding window over the 60-s windows of data, when preparing the WESAD dataset for our GANs, we technically double our training data. Since subjects offer differing numbers of training windows, the total amount of windows for each LOSO run depends on the current test subject. The ranges are n=[494,496] without and n=[995,1000] with 30-s sliding windows. We thus see n≈1000 as the number of windows for each DP-CGAN after leaving a test subject out for LOSO training. The number of unique windows, on the other hand, stays at n≈496 since the overlapping windows from sliding do not include new unique data points but instead just resample the already included points from the original 60-s windows. Thus, the original data points are only duplicated into the created intermediate sliding windows, meaning they are not unique anymore. To resolve this issue, we calculate the noise using the unique training set size of n≤496. We, however, take 2× the number of epochs, which translates to seeing each unique data point twice during training and accounts for our increased sampling probability for each data point. We subsequently choose $\delta =1\times 10^{-3}$ according to $\delta \ll \frac{1}{n}$ [16] and use a norm clip of C=1.0.

#### 4.8.2. For Classification Models

When training our three different classification models in the privacy-preserving setting, we only need to apply DP when including original WESAD data since the DP-CGANs already produce private synthetic data. In these cases, we mostly keep the same hyperparameters for training as before. We, however, exchange the Adam for the DP-Adam optimizer with the same learning rate from the Tensorflow Privacy library, which is an Adam version of DP-SGD. Regarding the DP noise, we calculate the needed amount conforming to the wanted guarantee before training. We already know the number of epochs and the batch size, which we also set for the microbatches. We, however, also have to consider other relevant parameters. The needed noise depends on the number of training samples, which for us is the number of windows. Since we do not use the 30-s sliding windows when training classifiers on the original WESAD data, all windows are unique. We find (at most) n≤496 remaining windows when omitting a test subject for LOSO training. This leads to $\delta =1\times 10^{-3}$ according to $\delta \ll \frac{1}{n}$ [16]. We finally choose a norm clip of C=1.0.

## 5. Results

In this section, we present the achieved results for our different evaluation criteria.

### 5.1. Synthetic Data Quality Results

This section summarizes the results of our analysis regarding the ability of our generated data to simulate original data. We give visual and statistical evaluations.

#### 5.1.1. Two-Dimensional Visualization

In Figure 5, we use PCA and t-SNE to visualize the multimodal signal profiles in a lower two-dimensional space. We give separate diagrams for each model and also differentiate between non-stress and stress data. PCA and the t-SNE visualizations both show how well the diversity of the original data distribution has been mimicked and whether synthetic data point clusters form outside of it or miss the outliers of the real data.

Except for missing some smaller outlier clusters, the CGAN and DP-CGAN at ε=1 visually seem to give a good representation of the original allocation. The CGAN shows to have a slight advantage in t-SNE as seen in Figure 5c,d, where the DP-CGAN (ε=1) gives a straighter line cluster and thereby misses the bordering zones of the point cloud.

The other GANs generally also show some clusters that mostly stay within the original data. However, they tend to show more and stricter separation from the original points. They also miss clusters and form bigger clusters than the original data in some locations. The DGAN shows an especially strict separation to the original cluster for the t-SNE stress data in Figure 5d, which induces problems when training with both data and might not correctly represent the original data.

In Figure 6, we examine how much each signal contributes to the two major PCs in our PCA model for the WESAD data. ACC shows significant importance in both non-stress and stress samples. TEMP also plays a role in both scenarios, particularly in non-stress. EDA contributes notably only in stress conditions, consistent with its role in stress detection. Conversely, BVP appears to have minimal impact on the PCs. Unlike PCA, t-SNE does not provide a direct interpretation of its dimensions, as they are a complex function of all features designed to preserve the local structure of the data.

#### 5.1.2. Signal Correlation and Distribution

Looking at the signal correlation matrices presented in Figure 7, the diagonal and upper triangle right of it plot the Pearson correlation coefficients between our signals. The main focus is to find correlations between the labeling of non-stress and stress with any of the signals. For the WESAD dataset, we mainly see a strong positive correlation from EDA and some already significantly lower but still visible negative correlation from TEMP. For the GANs, it is important to stay fairly close to this label correlation ratio to allow a good stress classification on their data. We can see that both EDA and TEMP ratios are caught well by the DGAN and CGAN data. This is also true for the rest of the correlation matrix with the CGAN being slightly more precise overall.

In the lower row, we see the DP-CGAN results, where the GANs at ε=10 and ε=1 are able to keep the highest correlation for EDA. We, however, also observe a clear over-correlation of BVP and also between multiple other signals when compared to the WESAD data. Thus, the overall quality is already reduced. Finally, comparing to the DP-CGAN at ε=0.1, we see that the model transitions away from EDA to instead focus on ACC and TEMP. The correlations between other signals are comparable to ε=10 and ε=1, but with losing the EDA ratio, the GAN at ε=0.1 loses its grip on the main correlation attributes.

Focusing on the lower half of the matrices to the left of the diagonal, we observe the corresponding *p*-values for our plotted correlations. Most correlations within the WESAD data are highly statistically significant, with *p*-values below 0.01. The ACC_x-Label correlation remains statistically significant with a *p*-value of 0.03. However, the BVP-Label correlation stands out with a *p*-value of 0.67, indicating no statistical significance. In our analysis of the GAN-generated data, we aim for a distribution that closely mirrors the original correlations. The CGAN closely matches the WESAD statistics, whereas other GAN models, such as the DGAN and DP-GANs at ε=10 and ε=1, predominantly show *p*-values of significance, failing to capture the BVP-Label and ACC_x-Label correlations. Conversely, the DP-GANs at ε=0.1 even add two different pairs with low significance. Still, all GANs are able to match the overall high statistical significance of their correlations.

We now take a closer look at the distribution density histogram of the EDA signal data in the GAN datasets compared to the original WESAD dataset in Figure 8. We picked EDA as our sample because of its strong correlation to the stress labeling and therefore significance for the classification task. The evaluation results for all modalities are available in Figure A1 of Appendix A. Comparing the distribution density in non-stress data, we can see how EDA in WESAD is mostly represented with very low values because of a large peak at zero and a clustering on the left end of the x-axis (x=[0.0,0.3]). While the DGAN and CGAN show similar precision, with only smaller deviations from the original distribution, we can see the DP-CGANs struggle with adhering to it in different ways. The DP-CGANs at ε=10 and ε=0.1 tend to overvalue EDA leading to a skewing of the distribution to the right on the x-axis. The DP-CGAN at ε=1, however, shows the opposite direction and a greatly underrepresented EDA by shifting further to the left and showing an extremely high density at x=0 that neglects the other values.

When comparing EDA distribution for stress, we instead observe a variety of values and a cluster located on the right half of the x-axis (x=[0.6,0.8]). Here, the CGAN clearly wins by delivering a good representation over all the values. The DGAN, on the other hand, shows a too-high distribution on the highest signal values (x=[0.9,1.0]). The private GANs at ε=10 and ε=1 generally show a good representation, which is only slightly shifted to favor lower values than in the original data. The DP-CGAN at ε=0.1 goes a bit too far in this direction by keeping a high density at x=0, leading to the worst representation of the general direction of higher EDA values for the original stress data.

#### 5.1.3. Indistinguishability

The results of our C2ST for indistinguishability are given in Table 3. Next to the generated data from our GAN models, we also include a test result on the original WESAD data that was not seen by the classifier, i.e., it is different than the WESAD data we hand to the classifier as real data. Creating synthetic data that come close to the original WESAD data would be the optimal case and thus the performance of our classifier in detecting such data as fake is the empirical lower bound achievable for our GANs. With this in mind, we can see that the CGAN not only has the best results but also comes close to the unseen WESAD data results, showing that the CGAN data are almost indistinguishable from real data. For the DP-CGANs, we see mixed success, where the classifier performs especially well in identifying our synthetic stress data but is fooled more by the non-stress data from the GANs at ε=1 and ε=0.1. DP-GAN data at ε=10 and DGAN data both seem to be an easy task for the classifier, which is able to clearly separate them from the original data.

### 5.2. Stress Detection Use Case Results

In this section, we report our results regarding an actual stress detection task. We first formulate a baseline in Section 5.2.1 to have a basic comparison point. We then present the results of our methods using deep learning and synthetic GAN data in Section 5.2.2.

#### 5.2.1. Baseline Approach

For creating a non-private baseline approach, we build a Logistic Regression (LR) model on the spectral power of our signals in the same LOSO evaluation setting. We consider each possible combination of signals as inputs for our LR to analyze their influence. Figure 9a presents the performance outcomes regarding all variations. The combination of BVP, EDA, and ACC_x yields the highest F1-score of 81.38%, while the best individual signal is EDA at 76.94%. Although being part of the best-performing set, BVP and ACC_x score only 28.07% and 0% on their own, respectively. Their weak results mainly highlight the crucial role of EDA in stress detection but also show that combining signals is critical in identifying further moments of stress that are not perfectly aligned with just EDA.

Figure 9b shows the coefficients of our LR model trained on all signals, indicating the significance of each feature. The coefficients describe how the probability of the model outcome changes with a change in the input variable when holding all other inputs constant. It thereby highlights the importance of an input feature on the outcome, i.e., for classifying the stress label. EDA is confirmed as the most influential feature, aligning with its strong association with stress. Although ACC_x is part of the best-performing combination, its impact is modest. BVP even displays minimal importance, despite its same presence in the optimal set.

To further study signal importance, we examine the differences in average spectral power between stress and non-stress data for our signals, as shown in Figure 9c. We use the average percentage change, which calculates the change between two values based on their average, allowing for comparisons without designating one value as the reference. Overall, the percentage change between stress and non-stress data is 13%; however, specific signals show a much larger gap. Notably, EDA exhibits a significant difference of 128%, with a considerably higher average spectral power under stress conditions. Conversely, TEMP shows a 39% higher average for non-stress conditions. While ACC_y and ACC_z display moderate changes, BVP and ACC_x show only minor differences.

Figure 9a–c each illustrate varying levels of importance for our signals, but consistently highlighting EDA as the most significant. The influence of other signals varies depending on the model and analytical tools used. In the LR model performance, BVP and ACC_x are prominent alongside EDA, yet BVP’s importance is diminished in the LR model’s coefficients. Conversely, spectral power analysis identifies TEMP as the second most crucial signal after EDA, with other signals showing only minor variations between stress and non-stress conditions. Also taking into account Figure 6, we can determine that while EDA is consistently crucial, the contribution of other signals can depend significantly on the specific analytical approach and model settings. This leads to a complex pre-requisite of signal analysis and selection in the stress detection task using basic tools like our baseline LR model. The approach based on deep learning models in the following section can help reduce the need for careful feature selection and evaluation through its ability to automatically extract and prioritize relevant features directly from the input data.

#### 5.2.2. Deep Learning Approach

We evaluate the usefulness of our synthetic data in a practical scenario of stress detection on the WESAD dataset. To enhance existing methods, we introduce synthetic GAN data into the training using our AUGM and TSTR settings, as described in Section 4.6. In Table 4, we give a full summarizing view of our results for both settings and take into account different amounts of synthetic data, as well as differing privacy levels.

On the WESAD dataset, our models perform well but not exceptionally well regarding the related work presented in Section 3, which could be due to the, in some aspects, differing data preparation we employed to train our GANs. The subsequently generated data inherently have the same processing and we thus also used them for the WESAD dataset to better combine the data in our stress detection evaluation. It seems like the stress classification models disagree with the GAN models, to some extent, in terms of how the data should be processed. This is especially true for the TSCT model, which stays behind the CNN and CNN-LSTM by a good portion. We can see, however, that the introduction of GAN data instead brings back the advantage of our pre-processing strategy, leading to stronger classification results on all privacy levels.

Another general trend is the CGAN outperforming the DGAN data, which is in line with the data quality results in Section 5.1. We further see that an increased number of synthetic subjects is not always better in performance since the datasets of 15 generated subjects and 100 subjects are placed closely together and exchange the crown between settings.

Comparing the AUGM and TSTR settings, we can see a clear favorite in the non-private setting at ε=∞. Here, the AUGM strategy using both the original WESAD and GAN data clearly outperforms our TSTR datasets with solely synthetic data. We achieve our best result of about 93% using AUGM with 100 subjects of the CGAN and using a CNN-LSTM model. The TSTR results still tell a story though. From the non-private TSTR, we can see the high quality of our synthetic data because we can already reach 91.33% without adding original WESAD data.

We observe a paradigm shift in the private settings of ε={10,1,0.1}, where the TSTR strategy using DP-CGANs reigns supreme over the AUGM approach. The main difference lies in the training setup, where TSTR induces the needed noise already in the DP-CGAN training process. The WESAD-based methods instead (also) have to rely on noise when training the classifier, which shows to be at a substantial disadvantage. While the CNN-LSTM holds good results for all privacy levels with just the WESAD dataset, the TSCT and CNN fail miserably. The AUGM method is able to lift their performance but stays significantly behind the TSTR results. TSTR takes the lead with results of 88.04% and 85.36% at ε=10 and ε=1, respectively. In both cases, we use 15 synthetic subjects and a CNN model. This changes for ε=0.1, where we achieve 84.19% using 100 subjects and a CNN-LSTM. The utility–privacy trade-off of our DP approach compared to the best non-private performance of 93.01% is ΔF1={−4.97%,−7.65%,−8.82%} for ε={10,1,0.1}, which can be considered a low utility loss especially for our stricter privacy budgets.

## 6. Discussion

The CGAN wins over the DGAN in our usefulness evaluation regarding an actual stress detection task conducted in Section 5.2.2. In non-private classification, we are, however, still unable to match the state-of-the-art results listed in Table 1 with just our synthetic CGAN data. In contrast, we are able to surpass them slightly by +0.45% at a 93.01% F1-score when combining the synthetic and original data in our AUGM setup using a CNN-LSTM. The TSCT model generally tends to underperform, while the performance of the CNN and CNN-LSTM models fluctuates, with each model outperforming the other depending on the specific setting. Our private classification models, which work best when only using synthetic data from DP-CGANs in the TSTR setting, show a favorable utility–privacy trade-off by keeping high performance for all privacy levels. With an F1-score of 84.19% at ε=0.1, our most private model still delivers usable performance with a loss of just −8.82% compared to the best non-private model, while also offering a very strict privacy guarantee. Compared to other private models from the related work presented in Table 1, we are able to give a substantial improvement in utility ranging from +11.90% at ε=10 to +14.10% at ε=1, and +15.48% at ε=0.1 regarding the F1-score. The related work on private stress detection further indicates a large number of failing models due to increasing noise when training with strict DP budgets [19]. We did not find any bad models when using our strategies supported by GAN data, making synthetic data a feasible solution to this problem. Our overall results in the privacy-preserving domain indicate that creating private synthetic data using DP-GANs before the actual training of a stress classifier is more effective than later applying DP in its training. Using just already privatized synthetic data is shown to be favorable because GANs seem to work better with the induced DP noise than the classification model itself.

In relation to our baseline results in Section 5.2.1, our method demonstrates a significant performance boost and the advantage of making the feature selection obsolete. Without additional GAN data, our non-private deep learning model delivers 86.48%, surpassing the baseline by 5.1%. The best non-private model incorporating synthetic data exhibits an even more substantial increase, outperforming the baseline by 11.63%. Moreover, our most private model at ε=0.1 still manages to outperform the best LR model by 2.81%. Overall, the deep learning approach, particularly when augmented with GAN data, proves to be superior to the baseline LR model.

Until now, we only consider the overall average performance from our LOSO evaluation runs; it is, however, also interesting to take a closer look at the actual per-subject results. In this way, we can identify if our synthetic data just boost the already well-recognized subjects or also enable better results for the otherwise poorly classified and thereby underrepresented subjects. In our results on the original WESAD data, we see that Subject ID14 and ID17 from the WESAD dataset are the hardest to classify correctly. In Table 5, we therefore give a concise overview of the results for the LOSO runs with Subject ID14 and ID17 as our test sets. We include the F1-scores delivered by our best synthetically enhanced models at each privacy level and compare them to the best result from the original WESAD data, as found in Table 4. We can see that our added synthetic data mostly allow for better generalization and improve the classification of difficult subjects. Even our DP-CGANs at ε=10 and ε=0.1, which are subject to a utility loss from DP, display increased scores. The other DP-CGAN at ε=10, however, struggles on Subject ID14. A complete rundown of each subject-based result for the selected models is given in Table A1 of Appendix A. The key insights from the full overview are that our GANs mostly facilitate enhancements in challenging subjects. However, especially non-private GANs somewhat equalize the performance across all subjects, which also leads to a decrease in performance in less challenging subjects. In contrast, private DP-CGANs tend to exhibit considerable differences between subjects, excelling in some while falling short in others. The observed inconsistency is linked to the DP-CGANs’ struggle to correctly learn the full distribution, a challenge exacerbated by the noise introduced through DP. Such inconsistencies may pose a potential constraint on the actual performance of our DP-CGANs on specific subjects.

While improving the classification task is our main objective, we also consider the quality of our synthetic data in Section 5.1. The CGAN shows to generate the best data for our use case, which are comparable to the original dataset in all data quality tests, while also performing best in classification. The DGAN achieves good results for most tested qualities but stays slightly behind the CGAN in all features and performs especially weakly in our indistinguishability test. We notice more and more reduced data quality from increasing DP guarantees in our DP-CGANs but still see huge improvements in utility for our private classification. Considering the benefits and limitations, the CGAN could potentially generate a dataset that closely approximates the original, offering a viable extension or alternative to the small WESAD dataset. The DP-CGANs, on the other hand, show their advantages only in classification but considering their added privacy attributes, the resulting data quality trade-off could still be tolerable depending on what the synthetic data are used for. The private data are shown to still be feasible for our use case of stress detection. For usage in applications outside of stress classification, e.g., other analyses in clinical or similar critical settings, however, the DP-CGAN data might already be too inaccurate.

Beyond the aforementioned points, our synthetic data approach, to a certain extent, inherits the limitations found in the original dataset it was trained on. Consequently, we encounter the same challenges that are inherent in the WESAD data. These include a small number of subjects, an uneven distribution of gender and age, and the specific characteristics of the study itself, such as the particular method used to trigger stress moments. With such small datasets, training GANs carries the risk of overfitting. However, we have mitigated this risk through the use of LOSO cross-validation. Further, as demonstrated in Table 5, our GANs have proven capable of enhancing performance on subjects who are underrepresented in earlier classification models. Nevertheless, questions remain regarding the generalizability of our stress classifiers to, e.g., subjects with other stressor profiles and the extent to which our GANs can help overcome the inherent shortcomings of the original WESAD dataset.

## 7. Conclusions

We present an approach for generating synthetic health sensor data to improve stress detection in wrist-worn wearables, applicable in both non-private and private training scenarios. Our models generate multimodal time-series sequences based on original data, encompassing both stress and non-stress periods. This allows for the substitution or augmentation of the original dataset when implementing machine learning algorithms. Given the significant privacy concerns associated with personal health data, our DP-compliant GAN models facilitate the creation of privatized data at various privacy levels, enabling privacy-aware usage. While our non-private classification results show only slight improvements over current state-of-the-art methods, our approach to include private synthetic data generation effectively manages the utility–privacy trade-offs inherent in DP training for privacy-preserving stress detection. We significantly improve upon the results found in related work, maintaining usable performance levels while ensuring privacy through strict DP budgets. Compared to the current basic anonymization techniques of metadata applied to smartwatch health data in practice, DP offers a provable privacy guarantee for each individual. This not only facilitates the development and deployment of accurate models across diverse user groups but also enhances research capabilities through the increased availability of public data. However, the generalizability of our classifiers to subject data with differing stressors, and the potential enhancement of these capabilities through our synthetic data, remain uncertain without additional public data for evaluation.

Our work sets the stage for how personal health data can be utilized in a secure and ethical manner. The exploration of fully private synthetic data as a viable replacement for real datasets, while maintaining utility, represents a promising direction for making the benefits of big data accessible without compromising individual privacy.

Looking ahead, the potential applications of our synthetic data generation techniques may extend beyond stress detection. They could be adapted for other health monitoring tasks such as heart rate variability, sleep quality assessment, or physical activity recognition, where privacy concerns are similarly demanding. Moreover, the integration of our synthetic data approach with other types of wearable sensors could open new avenues for comprehensive health monitoring systems that respect user privacy. Future work could also explore the scalability of our methods in larger, more diverse populations to further validate the robustness and applicability of the generated synthetic data.

## Figures and Tables

**Figure 1 sensors-24-03052-f001:**
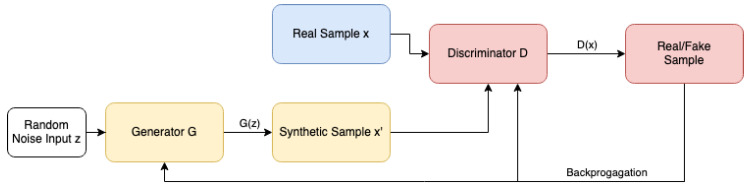
A brief description of the basic GAN architecture: The generator, denoted as *G*, creates an artificial sample $x^{\prime }$ using a random noise input *z*. These artificial samples $x^{\prime }$ and the real samples *x* are fed into the discriminator *D*, which categorizes each sample as either real or artificial. The classification results are used to compute the loss, which is then used to update both the generator and the discriminator through backpropagation.

**Figure 2 sensors-24-03052-f002:**
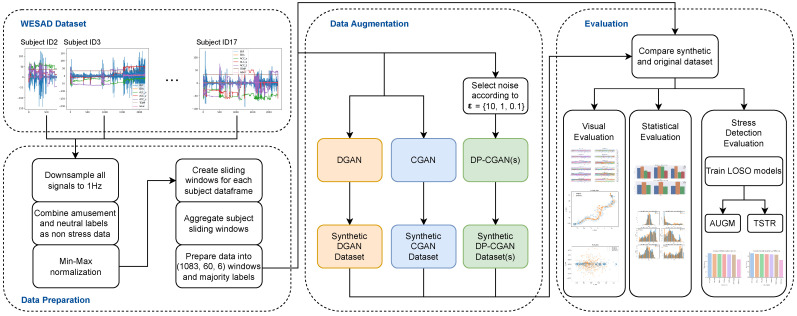
Our experimental methods are illustrated by the given workflow. In the first step, we load and pre-process the WESAD dataset. We then train different GAN models for our data augmentation purposes. Each resulting model generates synthetic data, which are evaluated on data quality and, finally, compared on their ability to improve our stress detection models.

**Figure 3 sensors-24-03052-f003:**
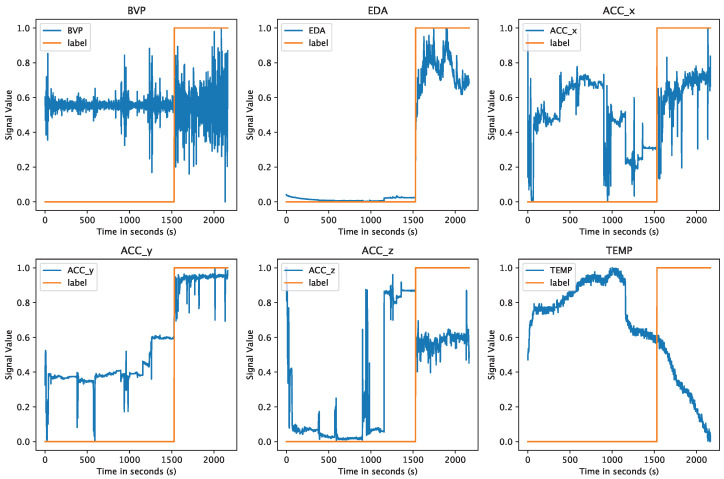
The individual signal modalities plotted for Subject ID4 after resampling, relabeling, and normalizing the data. The orange line shows the label, which equals 0 for non-stress and 1 for stress.

**Figure 4 sensors-24-03052-f004:**
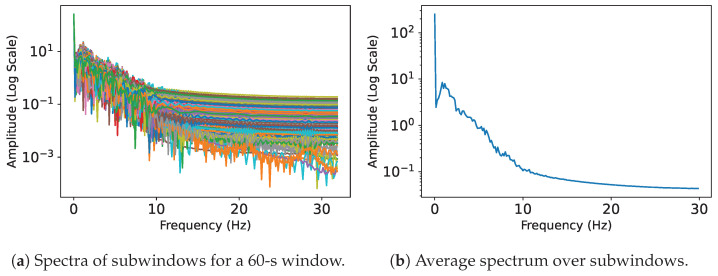
The spectrum plots from the FFT calculations of all subwindows in a 60-s window (**a**), and the plot of the averaged spectrum representation over these subwindows (**b**).

**Figure 5 sensors-24-03052-f005:**
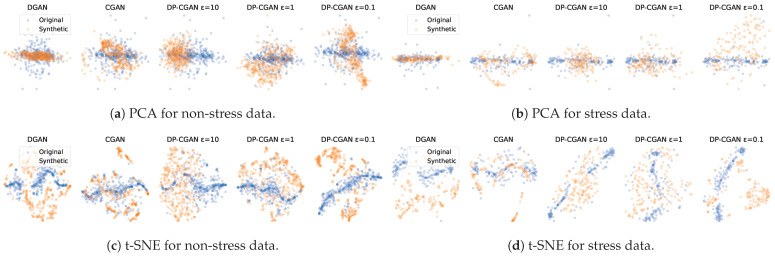
Visualization of synthetic data from our GANs using PCA and t-SNE to cluster data points against original WESAD data. Generated data are more realistic when they fit the original data points.

**Figure 6 sensors-24-03052-f006:**
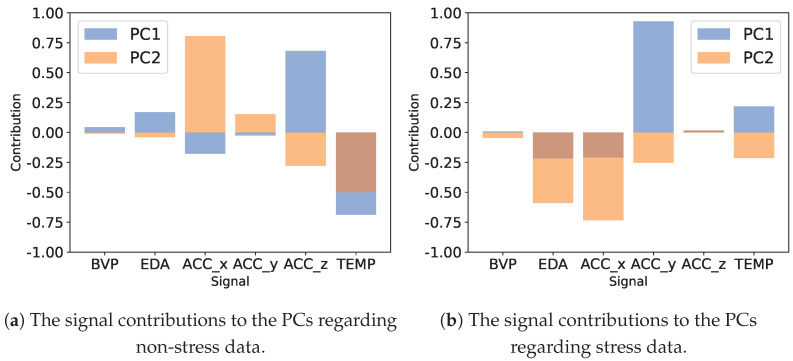
The signal contributions to the two PCs of our PCA model fitted on the original WESAD data. A high positive or negative contribution signifies that the feature greatly influences the variance explained by that component.

**Figure 7 sensors-24-03052-f007:**
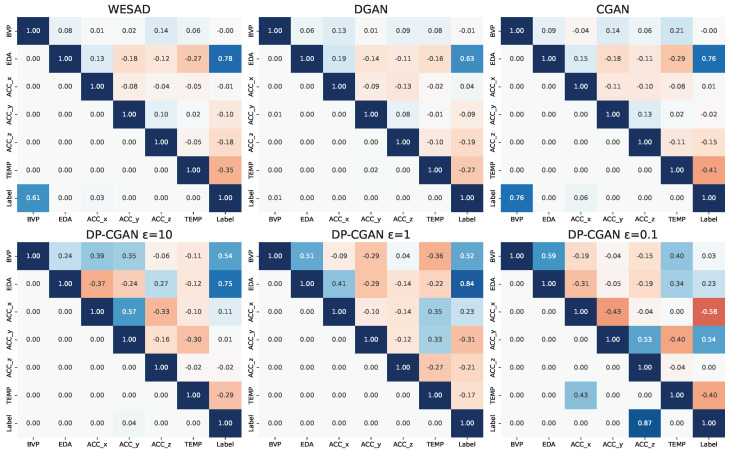
The matrices showing the Pearson correlation between the available signals. We compare real WESAD data and data from each of our GANs. In each matrix, the diagonal and all values to the right of it represent the correlation between signals. A higher value signifies a stronger correlation. The lower half of the matrices, left of the diagonal, shows the corresponding *p*-values for the signal correlation. A lower *p*-value translates to a higher statistical significance.

**Figure 8 sensors-24-03052-f008:**
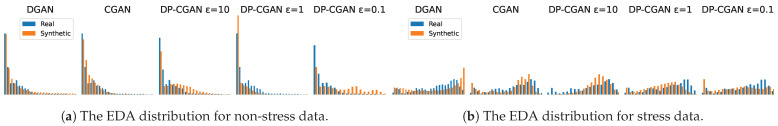
Histograms showing the distribution density of EDA signal values compared between original and generated data. The y-axis gives the density as y=[0,12], and on the x-axis, the normalized signal value is x=[0,1]. The plots for all signal modalities are located in Figure A1 of Appendix A.

**Figure 9 sensors-24-03052-f009:**
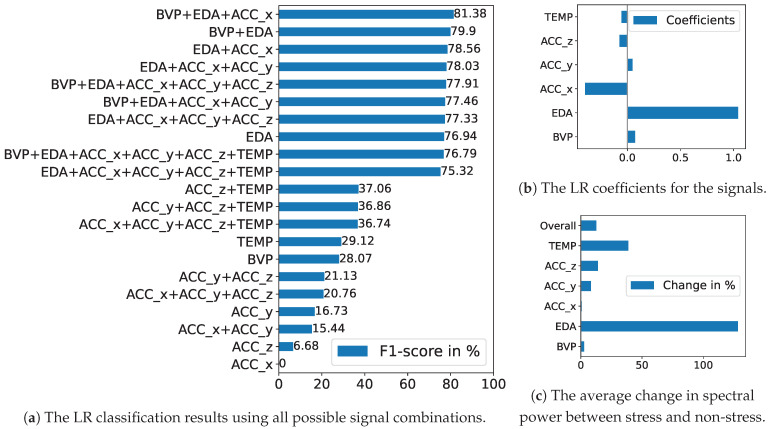
The results of our baseline experiment on stress detection using spectral power features. We employ a Logistic Regression (LR) model and test the effectiveness of various signal combinations.

**Table 1 sensors-24-03052-t001:** Performance results of relevant related work evaluated on WESAD dataset for modalities collected from wrist devices regarding binary (stress vs. non-stress) classification task. We compare accuracy (%) and F1-score (%) and include achieved ε-guarantee regarding DP.

Reference	Model	Data	Accuracy	F1-Score	Privacy Budget ε
[20]	RF	WESAD	87.12	84.11	*∞*
[21]	LDA	WESAD	87.40	N/A	*∞*
[22]	CNN	WESAD	92.70	92.55	*∞*
[19]	TSCT	WESAD	91.89	91.61	*∞*
Ours	CNN-LSTM	CGAN + WESAD	**92.98**	**93.01**	*∞*
[19]	DP-TSCT	WESAD	78.88	76.14	10
Ours	CNN	DP-CGAN	**88.08**	**88.04**	10
[19]	DP-TSCT	WESAD	78.16	71.26	1
Ours	CNN	DP-CGAN	**85.46**	**85.36**	1
[19]	DP-TSCT	WESAD	71.15	68.71	0.1
Ours	CNN-LSTM	DP-CGAN	**84.16**	**84.19**	0.1

**Table 2 sensors-24-03052-t002:** The subwindow length per signal depending on its frequency range and the resulting number of inputs for the classification model, as described by Gil-Martin et al. [22].

Signal	Frequency Range	Subwindow Length	# Inputs
ACC (x,y,z)	0–30 Hz	7 s	210
BVP	0–7 Hz	30 s	210
EDA	0–7 Hz	30 s	210
TEMP	0–6 Hz	35 s	210

**Table 3 sensors-24-03052-t003:** The results of the classifier two-sample test (C2ST), where a low accuracy closer to 0.5 is better. We also include the results of the unseen WESAD test data, which constitute an empirical lower bound.

	*WESAD (Unseen)*	DGAN	CGAN	DP-CGAN ε=10	DP-CGAN ε=1	DP-CGAN ε=0.1
Both	*0.59*	0.93	**0.61**	0.93	0.77	0.75
Stress	*0.72*	0.94	**0.77**	1.00	0.90	0.85
Non-stress	*0.70*	0.90	**0.71**	0.99	0.83	0.91

**Table 4 sensors-24-03052-t004:** A summarization of our stress classification results in a comparison of our strategies using synthetic data with the results using just the original WESAD data. We include different counts of generated subjects, privacy budgets, and classification models. Each setting is compared on the F1-score (%) as our utility metric.

Strategy	Dataset (s)	Subject Counts	Privacy Budget ε	TSCT	CNN	CNN-LSTM
Original	WESAD	15	*∞*	80.65	88.00	86.48
TSTR	DGAN	15	*∞*	80.60	85.89	85.33
TSTR	CGAN	15	*∞*	87.04	88.50	90.24
TSTR	DGAN	100	*∞*	73.90	84.46	79.31
TSTR	CGAN	100	*∞*	86.97	87.96	91.33
AUGM	DGAN + WESAD	15 + 15	*∞*	82.86	88.45	90.67
AUGM	CGAN + WESAD	15 + 15	*∞*	88.00	91.13	90.83
AUGM	DGAN + WESAD	100 + 15	*∞*	86.94	87.28	88.14
AUGM	CGAN + WESAD	100 + 15	*∞*	90.67	91.40	**93.01**
Original	WESAD	15	10	59.81	46.21	73.18
TSTR	DP-CGAN	15	10	87.55	**88.04**	84.84
TSTR	DP-CGAN	100	10	85.28	86.41	85.19
AUGM	DP-CGAN + WESAD	15 + 15	10	64.24	73.66	71.70
AUGM	DP-CGAN + WESAD	100 + 15	10	71.96	73.50	69.59
Original	WESAD	15	1	58.31	26.82	71.82
TSTR	DP-CGAN	15	1	82.90	**85.36**	78.07
TSTR	DP-CGAN	100	1	83.75	77.43	83.94
AUGM	DP-CGAN + WESAD	15 + 15	1	68.55	75.76	71.70
AUGM	DP-CGAN + WESAD	100 + 15	1	50.06	62.03	71.75
Original	WESAD	15	0.1	58.81	28.32	71.70
TSTR	DP-CGAN	15	0.1	76.27	81.35	76.53
TSTR	DP-CGAN	100	0.1	76.54	83.00	**84.19**
AUGM	DP-CGAN + WESAD	15 + 15	0.1	68.99	73.89	71.70
AUGM	DP-CGAN + WESAD	100 + 15	0.1	35.05	61.99	71.70

**Table 5 sensors-24-03052-t005:** LOSO results for Subject ID14 and ID17 from the WESAD dataset. We compare the achieved F1-scores (%) based on the original WESAD data and on the best synthetically enhanced models. The full coverage of all subject results is found in Table A1 of Appendix A.

WESAD Subject	WESAD	CGAN	CGAN + WESAD	DP-CGAN ε=10	DP-CGAN ε=1	DP-CGAN ε=0.1
ID14	54.46	74.88	**77.22**	69.44	61.00	57.22
ID17	53.57	**91.39**	88.61	65.18	43.04	83.33

## Data Availability

Publicly available datasets were analyzed in this study: https://ubicomp.eti.uni-siegen.de/home/datasets/icmi18/ (accessed on 8 May 2024) [20]. The implementations for the experiments in this work can be found here: https://github.com/luckyos-code/Privacy-Preserving-Smartwatch-Health-Data-Generation-Using-DP-GANs (accessed on 8 May 2024).

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
