# Peer review of "Generating Synthetic Health Sensor Data for Privacy-Preserving Wearable Stress Detection"

_sensors, 2024, doi:10.3390/s24103052_

Round 1

Reviewer 1 Report

Comments and Suggestions for Authors

The paper presents the latest health care application based studies. The topic is good the paper is well presented however I have following concerns and suggestions before further proceedings:

Some significant results can be mentioned in the abstract to make it more comprehensive.

The main method that used, can be more elaborated and justify that on what basis you select this method.

Please explain the Table 2 in the text and its significance as well.

For the related work, latest work should consider i.e. of 2023 and 2024. Some latest works related to the healthcare are ignored such as https://doi.org/10.1109/ACCESS.2023.3340884 and similar works.

How your method/model differentiated between the different ages and diseases based patient among so many people? Please justify

Some results that are explained in the text can be graphically explained.

Reviewer 2 Report

Comments and Suggestions for Authors

In this paper, the authors have trained a GAN on physiological measurements of stress (15 subjects). The paper is interesting, even though it is not clear to me whether a GAN is really necessary to obtain this result. As the authors note, there is a strong correlation between stress/non-stress and several predictor variables (so why do the effort of a GAN). Further, it is unclear how the spectral analysis - which seems to be the input of the GAN - is conducted. The authors first explain that the data is downsampled to 1 Hz and next select time windows of 250 ms, this is clearly incompatible and raises questions on the full processing pipeline. 

Major comments: 

  • EDA seems to be perfectly aligned with stress (see Figure 3). Why not stop here and simply use a cutoff on EDA to define stress? What would the accuracy be if only EDA was taking into account? Put differently, what is the added value of the other parameters? 
  • the usual problem with these approaches is the lack of available data. Yet, the authors downsample the few data points that are available to 1 Hz (which indeed speeds up training, but reduces the number of examples available). Further, they used a sliding window approach with windows of 60seconds that overlap 30 seconds. Why not increase the overlap, e.g. to 3 seconds to further increase the number of samples available for training? 
  • why do the authors consider the fact that “modalities correlate to their assigned label” an issue? Isn’t this what this is all about? Detecting stress from the labels? Without correlation, the results would be very poor, no? 
  • what are the intrinsic privacy concerns related to this kind of data? A subject’s response to stress is very unlikely to be unique - so - even if the original data (without metadata) - would be publicly available, how could one identify a single subject from this dataset? 
  • I am confused by the spectral calculation (section 4.7.1). Could the authors clarify on which final window they calculated the FFT? If it is only 0.25 seconds long, that corresponds to a frequency resolution of 4 Hz, so the spectrum is evaluated at 0, 4, 8, 12, … Hz => Why would this be a good idea? Why is it needed to take such a short time windows? And even more importantly, how do you actually do that with a signal that has been downsampled to 1 Hz. You have one data point per second, so how can your final time window be 0.25 seconds long? 
  • Was any windowing applied? Or averaging to increase SNR? 
  • Why did the authors resort to zero-padding? Can you explain why this was needed? 
  • what do the authors mean with “missing values in the higher frequencies” ? 
  • Can the authors plot a spectrum? 
  • Figure 4: what are the principal components? How do the different parameters contribute to the different PCs ? Is the PCA done for each of these subplots separately or is the same PCA used for all of these plots? 
  • isn’t the success of the GAN measured by the ability of the discriminator to guess? How well does it get? What accuracy does the discriminator achieve? And what does it tell us about the value of the generator? Why do the authors consider the other parameters to be relevant? (section 5.1.1 and 5.1.2). How should I interpret figure 5? 
  • What is the difference in average spectral power between stress and non-stress time windows? And what is the accuracy of having a very basic logistic regression with spectral power as input and output stress vs non-stress. This type of very basic analysis is required to provide a bench mark to which the final results obtained through the GAN can be compared. 

Minor comments: 

  • please first introduce abbreviations before using them
  • PCA could be better explained: please explain that the major PCs capture most of the variance 
  • what kind of filters were used on these time series? 
  • were any artefacts rejected (e.g. large movements) ? 
  • Section 5.1.2. correlation is obviously symmetric, why don’t you use the upper half to show the correlation coefficient (Pearson I assume that is) and the lower half of the matrix to show the p-values associated with this correlation (with a -log10 transformation for visualisation purposes)
  • Figure 6: can you clarify the distinction between the figures on the left and right-hand side of the figure? I now see it is below, but is rather unclear. 
  • how does the histogram look like for actual EDA data, split across stress and non-stress? 

Reviewer 3 Report

Comments and Suggestions for Authors

The manuscript, "Generating Synthetic Health Sensor Data for Privacy-PreservingWearable Stress Detection," is relevant. The research uses synthetic data generated by Generative Adversarial Networks (GANs) to detect stress in sensor data captured by smartwatches. The study emphasizes the importance of privacy preservation through Differential Privacy (DP).

Major suggestions:

In the introduction and conclusion sections of the paper, it is important to emphasize the work's main contribution. This includes not only the generation of synthetic data using GANs and the incorporation of differential privacy but also how these contributions are integrated to address the specific challenge of stress detection in health sensor data collected from smartwatches.
It would be helpful to have a more in-depth discussion on how the authors' approach differs or improves existing solutions, particularly about synthetic data quality, stress detection effectiveness, and privacy preservation, despite mentioning prior work.
Please expand the "Synthetic Data Quality Evaluation" section with additional details on ensuring that the synthetic data generated is a faithful and helpful representation of the actual data beyond just model performance metrics. Please consider including sensitivity analyses or specific case studies in this expansion.
It is worth exploring different GAN architectures and configurations to optimize data generation. This may include tuning hyperparameters, experimenting with different loss functions, or exploring GAN to improve the quality of synthetic data.

Minor suggestions:

Clear and justify all methodological steps, including specific GAN model parameters and structure.

Round 2

Reviewer 3 Report

Comments and Suggestions for Authors

The authors have adequately addressed my comments, and I now I think its ready for publication.